# Transient Electro-Thermal Coupled Modeling of Three-Phase Power MOSFET Inverter during Load Cycles

**DOI:** 10.3390/ma14185427

**Published:** 2021-09-19

**Authors:** Hsien-Chie Cheng, Siang-Yu Lin, Yan-Cheng Liu

**Affiliations:** 1Department of Aerospace and Systems Engineering, Feng Chia University, Taichung 407, Taiwan; ytalanben44556@gmail.com (S.-Y.L.); P0700100@o365.fcu.edu.tw (Y.-C.L.); 2Ph.D Program of Mechanical and Aeronautical Engineering, Feng Chia University, Taichung 407, Taiwan

**Keywords:** electro-thermal coupling analysis, power MOSFET inverter, power loss, circuit simulation, computational fluid dynamics, Foster thermal network

## Abstract

This study introduces an effective and efficient dynamic electro-thermal coupling analysis (ETCA) approach to explore the electro-thermal behavior of a three-phase power metal–oxide–semiconductor field-effect transistor (MOSFET) inverter for brushless direct current motor drive under natural and forced convection during a six-step operation. This coupling analysis integrates three-dimensional electromagnetic simulation for parasitic parameter extraction, simplified equivalent circuit simulation for power loss calculation, and a compact Foster thermal network model for junction temperature prediction, constructed through parametric transient computational fluid dynamics (CFD) thermal analysis. In the proposed ETCA approach, the interactions between the junction temperature and the power losses (conduction and switching losses) and between the parasitics and the switching transients and power losses are all accounted for. The proposed Foster thermal network model and ETCA approach are validated with the CFD thermal analysis and the standard ETCA approach, respectively. The analysis results demonstrate how the proposed models can be used as an effective and efficient means of analysis to characterize the system-level electro-thermal performance of a three-phase bridge inverter.

## 1. Introduction

Power electronics are widely used as powertrain components in the increasingly popular electric vehicles (EVs) and hybrid EVs, such as in electronic switches, converters, and inverters, to control and regulate electricity. In particular, three-phase voltage source inverters applied to control three-phase asynchronous induction motors are widely used in alternating current (AC) motor drives. Power semiconductors/modules inside inverters are the most crucial devices controlling the power conversion efficiency. In response to the urgent need for high-performance power conversion applications, the power semiconductor industry has recently seen rapid technological developments, such as insulated-gate bipolar transistors (IGBTs) [1,2], metal-oxide semiconductor field effect transistors (MOSFETs) [3,4], and even wide bandgap (WBG) silicon carbide (SiC) [5,6] and gallium nitride (GaN) power devices [7]. In contrast to IGBTs, MOSFETs comprise a number of advantageous features, such as a higher switching frequency and lower switching loss; accordingly, they have been used in a wide range of industrial applications, such as converters and inverters. 

Power devices unavoidably result in great losses in power during operation, including conduction losses resulting from the on-state resistance and switching losses stemming from simultaneous current and voltage waveforms and the influence of input/output capacitances and inductances. The trend for high power and downsizing in power devices is likely to bring about high power densities [8] and thus great power losses. Furthermore, a high power loss together with extreme operating conditions may potentially give rise to a high device junction temperature [6,7], which can cause various thermal and mechanical challenges, such as thermal instability and even unreliability in terms of thermal fatigue. For example, as a result of increased phonon concentration and lattice scattering, a high device junction temperature may lower the carrier mobility and thus raise the temperature-sensitive on-state resistance, which, in turn, increases the conduction loss and further elevates the device junction temperature. This process may, in the worst case, trigger thermal runaway reactions, ultimately leading to device breakdown. As well as this, a high device junction temperature can deteriorate the electrical performance and even be detrimental to the thermal–mechanical reliability of power devices (see, e.g., [5,9,10]). Hence, the temperature is one of the most important issues for power device applications. In order to ensure the safe and normal operation of power devices, the device junction temperature should be operated below the nominal rated temperature [11].

Pulse width modulation (PWM) three-phase bridge inverters are used in AC motor drive systems to convert the direct current (DC) power of batteries to a three-phase AC output with variable frequency and voltage for speed control. In conjunction with the high power density trend in power electronics, the wide variation in the frequency and phase current during load cycles can drive the device junction temperature beyond the temperature limit of power electronics, i.e., the maximum junction temperature rating, which would cause damage to or the failure of the inverters. Thus, there is a critical need for a more thorough comprehension of the thermal behavior of the power devices of inverters during operation. Before looking into the thermal issues of the inverters, a more in-depth understanding of their switching characteristics and power losses during load cycles is required. Several studies have reported that, in addition to supply voltage and gate resistance in the current loop, parasitic parameters are highly susceptible to the ringing and overshoots in the switching transients and, thus, can impact the switching loss [4,12,13]. For example, Cheng et al. [4] explored the switching characteristics and power losses of a silicon (Si) power MOSFET packaged in SOT-227 and a three-phase MOSFET bridge inverter during a switching operation in an effective compact circuit simulation model. They found that parasitic parameters have a considerable influence on the switching loss because of their effect on the switching waveform and speed. In addition to the parasitic effect, temperature also plays an important role in the switching and conduction losses of power devices (see, e.g., [1,14]). The device junction temperature during load cycles greatly influences the switching transients and power losses of power devices, which are, in turn, highly dependent on their device junction temperatures and parasitics. Thus, an accurate understanding of electromagnetic and dynamic electro-thermal (ET) coupled behaviors over a long-term operation is crucial for the safe operation of power components and systems.

According to previous studies from the literature, there have been extensive efforts heavily focused on the component-level exploration of the electromagnetic (EM), switching (power loss), and thermal behaviors (see, e.g., [2,15,16,17,18,19], but very limited work has been done on the system level, such as on three-phase bridge inverters. Heat generation in a three-phase inverter fluctuates at two widely different frequencies: the load current modulation frequency at the level of tens to hundreds of Hz and the switching frequency at the level of 10–100 kHz, where the switching time is only about a few hundred nanoseconds. Precisely modeling the PWM and switching events so as to thoroughly capture the switching transients and power losses of the three-phase inverter during the six-step operation requires an extremely small time step and thus enormous computing time. The problem becomes even more severe for high-operation-frequency applications. Moreover, the device junction temperature generally needs between hundreds and thousands of seconds to reach a steady state, depending on the thermal time constant. Directly coupling the electrical circuit analysis with three-dimensional (3D) transient computational fluid dynamics (CFD) thermal analysis to calculate the ET coupled behavior presents a great computational challenge because of limitations on storage space and computational power. In the literature, the problem has been successfully eased using resistance–capacitance (RC) thermal networks [20,21,22], such as Foster and Cauer networks, instead of directly carrying out the CFD thermal analysis. RC thermal networks can be an effective and favorable means for junction temperature estimation due to their unparalleled computational efficiency and flexibility for both thermal and electrical models [22]. A direct coupling of the detailed circuit simulation model and an RC thermal network model forms the so-called standard ET coupling analysis (ETCA) approach [23,24,25,26,27]. The standard ETCA approach still cannot fully address the circuit simulation difficulty in the two widely different frequencies, which even makes it impossible to explore the ET coupled behavior of the three-phase PWM inverter in a long-term operation. Accordingly, a more effective approach that can ameliorate the circuit simulation difficulty is critical needed. Reichl et al. [23] attempted to improve the computational efficiency of the standard ETCA approach using a four-step iterative process and an average dissipated power over an electrical cycle. Later on, Reichl et al. [28] alternatively presented a full 3D multilayer and multichip thermal component model with asymmetrical power distributions for dynamic ET simulation, where the 3D heat conduction equation is solved using finite difference methods, and the thermal component model is parameterized in terms of structural and material properties to facilitate the development of a library of component models for any available power module. It has been found, however, that the circuit simulation difficulty still cannot be removed. Accordingly, this study proposes a more effective and efficient dynamic ETCA approach, in which a simplified equivalent circuit simulation model is developed and fully coupled with a Foster thermal network model to account for the effect of the instantaneous junction temperature on the instantaneous power losses (switching and conduction). The proposed, simplified equivalent circuit simulation model can address the computational difficulty associated with the two significantly different frequencies and, therefore, can greatly reduce the computational cost and make the multi-temporal and long-term ETCA of a power conversion system much more feasible. In addition, to address the effects of parasitics on the switching transients and power losses, the proposed ETCA approach can be integrated with a 3D EM model. The proposed ETCA approach is demonstrated through the estimation of the ET coupled behavior of a voltage source three-phase bridge MOSFET inverter (see Figure 1) for brushless DC (BLDC) motor drive under natural and forced convection during a six-step operation. The established Foster thermal network model and the proposed ETCA approach are validated using CFD thermal analysis and the standard ETCA approach, respectively. 

## 2. Three-Phase MOSFET Bridge Inverter

The voltage source three-phase bridge inverter, which transforms DC power from a DC source into AC power for an AC load, is shown in Figure 1a. It comprises three parallel legs for phases a, b, c, and each of them contains two semiconductor switches (100 V and 350 A SOT-227 power MOSFET modules, as illustrated in Figure 1b): one at the upper side and the other at the lower side. These two switches in each leg are complimentarily operated. In total, there are six switches (S1–S6) in the inverter to create a three-phase bridge circuit with six switching arms that turn the current on and off, as displayed in Figure 2a. In detail, three of these six switches (S1, S3, and S5) are connected to a high-voltage-side DC voltage (hereinafter referred to as “upper-side switches”) and the others (S2, S4, and S6) to a low-voltage one (hereinafter referred to as “lower-side switches”). These arms are linked to each other through a connection bridge. In each modulation cycle, there is an electrical cycle (360°) with six switching steps, each with a duration of 60°, creating a cyclic three-phase pattern, as depicted in Figure 2b. At any commutation sequence in the six-step commutation logic, only one upper switch and one lower switch are turned on to energize two motor phase windings. The upper-side switches’ switching signals are kept discontinuously “on” (i.e., PWM “on”) with a duty cycle whereas the lower-side switches’ switching signals are always continuously “on” [29]. Thus, the upper switches are, alternatively, termed PWM power MOSFETs. The conduction sequence of one six-step commutation cycle is S1S4–S1S6–S3S6–S3S2–S5S2–S5S4, and the corresponding current states are ab, ac, bc, ba, ca, and cb. 

In order to enhance the current rating [14,30], three Si power MOSFET chips connected in parallel are embedded in the power MOSFET module. When controlling the PWM power modules, the common rectangular-wave PWM (RWPWM) technique is employed to generate a square-wave pulse via a signal generator, and a microcontroller is used to supply the gate pulses to these semiconductor switches. The current supplied to the power MOSFET modules is PWM-regulated through the rapid switching on and off of these switches. The ratio of the pulse width to the total signal period is defined as the duty cycle (D). When D = 50%, it is a square wave PWM (SWPWM). An increased duty cycle raises the electrical power supply to the semiconductor devices. The temperature-dependent on-state resistance and the output, transfer, and body diode characteristics of the power MOSFET module provided in the manufacturer’s datasheet and also in [4] are presented in Figure 3. Figure 3a,d reveal that the I–V characteristics of the power MOSFET and body diode show a strong temperature coefficient.

To facilitate heat dissipation, these six SOT-227 power MOSFET modules are bonded onto a thick heat spreader made of aluminum (Al) metal. The power MOSFET module contains one gate, one drain, and two source terminals for electrical connection. In addition, it is primarily composed of three Si power MOSFET chips; an Al_2_O_3_-based direct bonded copper (DBC) substrate; Al bond wires; bond pads made of Al metal; a Cu base plate; Cu terminal leads; three Sn-3.0Ag-0.5Cu (SAC305) solder layers for the bonding between the Si power MOSFET chips and the Cu terminal leads, between the Cu terminal leads and the DBC substrate, and between the DBC substrate and the Cu base plate; a polyphenylene sulfide (PPS) housing; and a quick-drying rubber-based adhesive applied to fill the cavity between the housing and the DBC/Cu terminal leads. The power MOSFET chips, DBC substrate, terminal leads, pads, and base plate have thicknesses of 0.33, 0.45, 0.8, 0.01, and 2.0 (mm). The thicknesses of the three solder layers are 0.05, 0.1, and 0.1 (mm). In total, there are twelve Al wires with the same lengths and cross-sectional areas on the Al pads of these three power MOSFET chips. 

## 3. Power Loss Prediction

The main types of power loss generated from power MOSFETs during operation include conduction, switching, and current leakage losses and diode conduction and reverse recovery losses. The leakage current loss is typically much lower than the conduction loss at low junction temperatures [14] and thus can be negligible if the junction temperature is appropriately controlled. The estimation of the conduction loss and switching loss of power MOSFETs, i.e., $P_{C}$ and $P_{S}$, during operation can be briefly demonstrated in the following. When power MOSFETs are switched on by the gate voltage, drain-source current flows across the resistive components, causing Joule heating and resulting in heat conduction loss. For a particular switching period, the conduction loss can be calculated from the drain-source current $I_{ds}$, on-state resistance $R_{ds(on)}$, and duty cycle *D* as
(1)$$P_{C}=\frac{D}{t_{s}}\int_{0}^{t_{s}}I_{ds}^{2}(t)R_{ds(on)}(T)dt$$

Since the on-state resistance has a large and positive temperature correlation, as seen in Figure 3c, the conduction loss is a strong function of temperature. For modeling simplicity, an average power loss is generally utilized in computation through the application of a root-mean-square (RMS) average current (*I_RMS_*) during a PWM operation. For an SWPWM control technique, *I_rms_* is denoted as
(2)$$I_{rms}=I_{ds}\sqrt{D}$$

With the RMS average current, the corresponding conduction loss can be expressed as
(3)$$P_{C}=I_{rms}^{2}R_{ds(on)}(T)$$

As a result of the simultaneous rise in current, from the leakage current to the on-state current $I_{DS}$, and fall in voltage, from the off-state voltage to the on-state voltage, power devices can induce considerable switching loss. Moreover, the PWM switching frequency has a positive and almost linear effect on the switching loss. A higher switching frequency causes a greater switching loss. As mentioned earlier, in addition to the device parameters, reverse recovery current, and gate drive current, the parasitic effect plays a significant role in the switching loss. Figure 4 shows typical voltage and current transients during turn-on and turn-off periods, where $V_{gs}$ is the gate-source voltage; $V_{TH}$ is the threshold voltage; $V_{gp}$ is the gate-plateau voltage; $V_{DD}$ is the supply voltage; $I_{peak}$ is the current spike (overshoot); $V_{ON}$ is the conduction voltage, which is equal to $I_{DS}R_{DS(on)}$; *V_GS_* is the gate drive voltage; and $V_{spike}$ is the voltage spike. The time increments $t_{2}-t_{1}$ and $t_{6}-t_{5}$ are defined as the rise time $t_{ir}$ and fall time $t_{if}$ of the on-state current $I_{ds}^{}$, respectively, and the time increments $t_{3}-t_{2}$ and $t_{5}-t_{4}$ are defined as the fall time $t_{vf}$ and rise time $t_{vr}$ of the drain-source voltage $V_{ds}$. Accordingly, the turn-on switching period $t_{s}^{on}$ is equal to $t_{3}-t_{1}$, and the turn-off switching period $t_{s}^{off}$ is equal to $t_{6}-t_{4}$. These switching transients are largely determined by parasitic parameters, such as the gate-drain capacitance $C_{gd}$, the gate-source capacitance $C_{gs}$, the drain-source capacitance $C_{ds}$, the drain inductance *L_d_*, the gate inductance *L_g_*, and the source inductance *L_s_*. These parasitic capacitances are closely related to the input capacitance *C_iss_* ($C_{iss}=C_{gs}+C_{gd}$), output capacitance *C_oss_* ($C_{oss}=C_{gd}+C_{ds}$), and reverse transfer capacitance *C_rss_* ($C_{rss}=C_{gd}$). Basically, they somewhat vary with the drain-source voltage *V_ds_*, as shown in Figure 5. Finally, the switching energy loss $E_{S}$ during a switching cycle is given as
(4)$$E_{S}=E_{on}+E_{off}=\int_{0}^{t_{s}^{on}}V_{ds}(t)I_{ds}(t)dt+\int_{0}^{t_{s}^{off}}V_{ds}I_{ds}(t)dt$$

The body diode can also contribute to the conduction loss and reverse recovery power loss. The former is produced when the upper switches (i.e., PWM power MOSFETs) are switched off and the current passes via the complementary lower switches (i.e., freewheeling diodes (FWDs)) [31]. The body diode conduction loss $P_{C}^{BD}$ across the switching period $t_{S}$ can be written as
(5)$$P_{C}^{BD}=\frac{1}{t_{S}}\int_{0}^{t_{S}}\left(V_{BD}^{0}I_{BD}(t)+R_{BD}(t)I_{BD}{}^{2}(t)\right)dt$$
where $I_{BD}$ is the current passing through the body diodes, $V_{BD}$ is the voltage of the body diodes, and $V_{BD}^{0}$ and $R_{BD}$ are the on-state zero-current voltage and resistance of the body diodes, respectively, which can be read from the diagrams in the package datasheet. Furthermore, when the body diodes are switched off, the charge stored in the drain-source capacitor of the FWDs must be released. The reverse recovery current is absorbed by the PWM power MOSFTs when they are switched on again. In fact, the reverse recovery effect is included in the power loss calculation for the upper-side switches that are turned on.

## 4. EM Electro-Thermal Analysis

### 4.1. EM Modeling

Maxwell’s equations, consisting of a set of coupled partial differential equations, are generally used to depict macroscopic electromagnetism phenomena. The equations indicate that EM waves moving along a field depend on time, space, the electric field, and the magnetic field [32]:(6)$$\nabla \cdot D=\bar{\rho }$$
(7)∇⋅B=0
(8)$$\nabla \times E=-\frac{\partial B}{\partial t}$$
(9)$$\nabla \times H=\bar{J}+\frac{\partial D}{\partial t}$$
where *D* denotes the electric displacement field or electric flux density, *B* the magnetic field density, *E* the electric field, $\bar{\rho }$ the free charge density (not including the bound charge), *H* the magnetic field intensity, and $\bar{J}$ the free current density (not including the bound current). Equations (1)–(4) are called Gauss’s law, Gauss’s law for magnetism, the Maxwell–Faraday equation, and the Ampère circuital law. The Ampère circuital law is also known as the Maxwell–Ampère law. The left-hand side of the Ampère circuital law possesses zero divergence due to the div–curl identity. Further expanding the divergence of the right-hand side, exchanging the derivatives, and applying Gauss’s law yields:(10)$$0=\nabla \cdot \left(\nabla \times H\right)=\nabla \cdot \bar{J}+\nabla \cdot \frac{\partial D}{\partial t}$$

This leads to
(11)$$\nabla \cdot \bar{J}=-\frac{\partial \bar{\rho }}{\partial t}$$

The free charge density does not vary with time (i.e., $\partial \bar{\rho }/\partial t=0$) for a stable current, and thus Equation (11) can be re-expressed as
(12)$$\nabla \cdot \bar{J}=0$$

Note that $\bar{J}=\sigma E$ and E=−∇V based on Ohm’s law. If the conductivity σ of the conductor material is assumed to be constant and evenly distributed, the equation governing the steady-state electric field can be derived as
(13)$$\nabla^{2}V=0$$

### 4.2. CFD Modeling

The mass, momentum, and energy conservation laws are solved in the CFD analysis using finite volume method. The conservation equations, namely mass, momentum, and thermal energy, in the Cartesian coordinate system under the assumption of Newtonian, incompressible, and steady fluid can be described as
(14)∇⋅v=0
(15)$$\rho \frac{\mathrm{Dv}}{Dt}=-\nabla p+\mu \nabla^{2}v+\rho g$$
(16)$$\rho \frac{De}{Dt}=-p\nabla \cdot v+\nabla \cdot (k\nabla T)+\Phi$$

In the above equations, v is the velocity; D/Dt=∂/∂t+(v⋅∇), the so-called material derivative; p is the pressure; ρ is the density; μ is the viscosity; g is the gravity; *T* is the temperature; *k* is the thermal conductivity; e is the internal energy; and Φ is the dissipation function, defined as
(17)$$\Phi =\nabla \cdot (\tau_{ij}\cdot v)-(\nabla \cdot \tau_{ij})\cdot v=\tau_{ij}\frac{\partial v_{i}}{\partial x_{j}}$$
where $\tau_{ij}$ is the viscous stress component
(18)$$\tau_{ij}=\mu \left(\frac{\partial v_{i}}{\partial x_{j}}+\frac{\partial v_{j}}{\partial x_{i}}-\frac{2}{3}\frac{\partial v_{k}}{\partial x_{k}}\delta_{ij}\right)$$

The body-force term in the Navier–Stokes equation, i.e., ρg, can be neglected for natural convection.

### 4.3. Foster Thermal Network Model

For a multiple-chip power system containing *n* power semiconductor devices, these devices will be subjected to temperature rise due to self-heating and cross-heating effects. More specifically, any chip in the module with a power dissipation *P* will undergo self-heating, causing a junction temperature rise *T_j_*, whereas the other devices will experience cross-heating, likewise leading to junction temperature elevation. In this work, a compact RC thermal network model in the form of a Foster network is applied for quick thermal simulation and easy implementation. The Foster network comprises a number of RC elements, where *R* is the thermal resistance (K/W) and is *C* the thermal capacitance (J/K). The Foster thermal network model does not have any physical meaning or represent the physical structure of power devices. In order to develop a Foster thermal network, it is necessary to obtain the transient thermal impedance curves for both the self- and cross-heating responses. In the transient thermal characterization, the thermal impedance *Z(t)* at a time *t* is used to determine the temperature variations ΔT(t)
(19)$$Z(t)=\frac{\Delta T\left(t\right)}{P(t)}=\frac{T_{j}(t)-T_{a}}{P(t)}$$

Using a Foster RC model, the above time-dependent thermal impedance *Z(t)* can be described as
(20)$$Z(t)=\sum_{i=1}^{n}R_{i}\left(1-\exp(-\frac{t}{\tau_{i}})\right)$$
where $\tau_{i}(i=1,\ldots ,n)$ are the *i*-th time constants, equivalent to the product of *R_i_C_i_* in the Foster network. For the three-phase inverter, consisting of six switching devices, the value of *n* is 6. The thermal impedance matrix of the three-phase inverter is shown below
(21)$$\left\{\begin{array}{c} T_{1}(t)\\ \vdots \\ T_{n}(t) \end{array}\right\}=\left[\begin{array}{ccc} Z_{11}(t) & \cdots & Z_{1n}(t)\\ \vdots & \ddots & \vdots \\ Z_{n1}(t) & \cdots & Z_{nn}(t) \end{array}\right]\left\{\begin{array}{c} P_{1}(t)\\ \vdots \\ P_{n}(t) \end{array}\right\}+\left\{\begin{array}{c} T_{a}\\ \vdots \\ T_{a} \end{array}\right\}$$
where *T_a_* is the ambient temperature. In the thermal impedance matrix, the diagonal components, namely *Z_ii_*, denote the self-heating impedance of the *i*-th switching device and the off-diagonal components, namely $Z_{ij}(i\neq j)$, stand for the cross-impedance between the *i*-th and *j*-th switching devices. The thermal impedance matrix can be established by applying a power step to the switching devices one by one and then measuring the corresponding temperature responses of each of them. 

In this work, the CFD code ANSYS Icepak (ANSYS Icepak 2020R2, Canonsburg, PA, USA) was used for the transient heat transfer simulation. The ANSYS Icepak CFD 3D model of the three-phase inverter is presented in Figure 6. The initial power at time zero (*t* = 0) was set to the estimated total power loss of the inverter at room temperature *T_a_*. Subsequently, curve fits of the simulated transient heating curves were performed to identify the parameters (i.e., *R* and *C*) and thus produce RC networks for all six of the power MOSFET switching devices in the inverter, with which the time-dependent thermal impedance matrix, as listed in Equation (21), was built. Using the characterized time-dependent thermal impedance matrix, the junction temperatures of these switching devices can be simply estimated with given power losses. In fact, this approach implies limitations. For example, the thermal model is established based on a linear system assumption, and the accuracy of the prediction actually relies on the degrees of nonlinearity, such as convection, radiation, and temperature-dependent material nonlinearity. 

## 5. Electro-Thermal Coupling Analysis (ETCA)

The analysis flow of the proposed ETCA platform is shown in Figure 7 and comprises three analysis layers: EM modeling, electrical simulation, and thermal analysis based on an RC thermal network model. In order to account for the temperature effect on the switching transients and even power losses (conduction and switching), the latter two analysis layers, i.e., electrical simulation and thermal analysis, are fully coupled to co-simulate the ET coupled behavior of the three-phase power MOSFET inverter. In the switching loss estimation, the parasitic capacitances are also considered V_ds_-dependent. 

In the platform, the ETCA starts with the parasitic extraction (inductances) using ANSYS^®^ Q3D Extractor, which is followed by the CFD thermal analysis and the fitting of the simulated heating curves in the time domain to establish the Foster thermal network model. ANSYS Icepak CFD software is responsible for solving the thermal problems in natural convection or forced convection and for deriving the transient thermal impedance curves. Instead of directly and iteratively performing the CFD analysis of natural or forced convection, the developed Foster thermal network model allows a rapid estimation of the junction temperature with different power conditions. Subsequently, with the characterized parasitic inductances together with the package model, including the output and transfer characteristics of the power MOSFET device, the diode characteristics, and the Vds-dependent parasitic capacitances, a detailed circuit simulation model of the three-phase inverter can be developed using ANSYS Simplorer to predict the switching transients and switching loss during the six-step operation. The detailed circuit simulation model of the three-phase inverter, together with the parasitic parameters (inductances) to be determined, is shown in Figure 2a. 

The proposed ETCA approach can be applied to improve the computational efficiency of standard ETCA. In addition to the Foster thermal network model, it incorporates a simplified equivalent circuit model, as shown in Figure 7b, where the inverter switches (S1–S6) are simply modeled by resistors. The temperature-dependent equivalent electrical resistances of the resistors (R1–R6) are used to simulate the temperature dependence of the corresponding power losses (P1–P6) of the inverter switches during the six-step operation. The power loss of each of these inverter switches is composed of the conduction and switching losses of the power MOSFET modules and the conduction loss of the body diodes. Once the power loss–temperature relationships of these resistors are known, the power losses of each of these inverter switches at any temperature can be readily determined, which suggests that there is no longer a need to perform a tedious and complex detailed circuit simulation to predict the temperature-dependent power losses. The established power loss–temperature relationships of these resistors are implemented in the simplified equivalent circuit model. The interactions between the Foster thermal network model and the simplified equivalent circuit model, which exchanges the power and temperature data, are fulfilled through ANSYS Simplorer as the linking layer. It is important to note that for the common 120-degree square-wave commutation, each inverter switch conducts for 120 electrical degrees in each periodic cycle, indicating that the inverter switch is turned off in the rest of the periodic cycle. The calculated power losses of these power switches during the 120 electrical degrees are averaged across the periodic cycle. In this work, the temperature-dependent power losses of these power switches during one PWM six-step commutation cycle are derived using the abovementioned detailed circuit model under different temperature conditions, and with these the equivalent electrical resistance–temperature relationship can be determined based on Ohm’s law.

## 6. Results and Discussion

### 6.1. Construction of Foster Thermal Network Model

Transient CFD thermal analysis of the three-phase inverter under natural convection was carried out using ANSYS Icepak. Then, constant power levels were sequentially set for each of the six switches, constituting six different power conditions. Accordingly, six parametric transient CFD analyses under natural convection associated with these six power conditions were performed using ANSYS Icepak and the corresponding transient junction temperature history profiles were collected. These temperature history profiles were further converted into transient thermal impedance curves. Two examples of the transient thermal impedance curves associated with Z_1*i*_(t) and Z_2*i*_(t) (*i* = 1, … 6) are presented in Figure 8. Subsequently, these transient thermal impedance curves were used to extract the corresponding parameters in Equation (21), namely the time constants and resistances, by curve fitting in the time domain. The fit of the least squares regression analysis was outstanding, with a calculated multiple determination coefficient over 0.998, suggesting that the variation in the thermal impedance data was well-explained. Two examples of the curve-fitted values of these parameters associated with the transient thermal impedance curves Z_1*i*_(t) and Z_2*i*_(t) (*i* = 1, … 6) shown in Figure 8 are presented in Table 1. According to Equation (21), these 36 time-dependent thermal impedance elements form the thermal impedance matrix, which was used to predict the junction temperatures of the power MOSFET chips under natural convection during load cycles.

The feasibility of the developed Foster network thermal model based on the linear system assumption was demonstrated by comparing it with the CFD thermal analysis results associated with these six inverter switches (S_1_–S_6_) obtained using ANSYS Icepak at two different power settings, i.e., [13.2, 13.2, 13.2, 20.1, 20.1, 20.1] (W) with a total power (P^T^) of 99.9 W and [11.4, 13.2, 15.9, 21.3, 24, 18] (W) with a total power of 103.8 W. The steady-state thermal analysis results are shown in Table 2. Note that the total power of the first power setting, i.e., *P**^T^* = 99.9 W, was the same as the initial preset power level used in the construction of the Foster thermal network model, while that of the second power setting (*P^T^* = 103.8 W) was about 4% or 3.9 W larger than the initial preset power level. It can be clearly seen that for the first power setting, the developed Foster thermal network model produced a result that was very consistent with the CFD thermal analysis. By contrast, for the second power setting, there was a maximum deviation of 3% from the result of the CFD thermal analysis. If the discrepancy is over 5%, the Foster thermal network model may need to be updated or re-established for better accuracy, according to the power loss presented during the ETCA analysis. In other words, as long as the total power of applied power settings is similar to that used to create the Foster thermal network model, the derived result should be sufficiently accurate.

### 6.2. ECTA Analysis of Three-Phase Inverter

The frequency-dependent parasitic parameters of the power MOSFET module and the three-phase inverter in a frequency sweep were explored using ANSYS^®^ Q3D 3D quasi-static EM field solvers with various assigned conducting nets. In this parasitic analysis, three conducting nets were defined to describe the current paths of the power MOSFET module, i.e., drain, source, and gate (i.e., *L_d_*, *L_g_* and *L_s_*), and ten conducting nets were assigned for the three-phase inverter in accordance with the switching sequence of the three-phase inverter, i.e., *L*_1_–*L*_7_ and *L*_10_–*L*_12_, as shown in Figure 2a. In the figure, *L*_8_ and *L*_9_ denote the drain and source inductances (*L_d_* and *L_s_*) of the power MOSFET module, respectively. It is worth mentioning that *L_s_* represents the sum of the parasitic inductances of the source terminal leads and Al wires. Furthermore, the three-phase load is modeled as a resistor (*R*)–inductor (*L*) series impedance, i.e., *R_a_-L_a_*, *R_b_-L_b_*, and *R_c_-L_c_*_,_ in Figure 2a. The parasitic inductances of the power MOSFET module extracted from the preceding inductive double-pulse test (DPT) circuit simulation at the working frequency of 20 kHz were 8.60, 5.47, and 7.53 nH and were associated with the gate, drain, and source terminals. As mentioned above, the source inductance is the sum of the parasitic inductances of the source terminal leads (i.e., 5.92 nH) and Al wires (i.e., 1.61 nH). The parasitic inductances associated with *L*_1_–*L*_7_ and *L*_10_–*L*_12_ were calculated in the authors’ previous work [4], and they are 23.34, 14.74, 25.52, 31.31, 6.93, 3.67, 54.89, 19.79, 19.52, and 19.78 (nH). These parasitic inductances, together with the package model (the output and transfer characteristics), the diode characteristics, and the V_ds_-dependent parasitic capacitances, were applied in the detailed circuit simulation model, with which, together with the Foster thermal network model, the standard ETCA approach was constructed. The load condition of the inverter was a power supply voltage of 50 V, an SWPWM (D = 50%) switching frequency of 10 kHz, and an output frequency of 55 Hz. The inductance and resistance for these three-phase loads were 20 µH and 0.125 Ω, respectively. In addition, the switching frequency, gate resistance *R_g_*, gate voltage *V_g_*, gate inductance *L_g_*, inductive load, and resistive load were set to 10 kHz, 1.6 Ω, 10 V, 8.6 nH, 20 μH, and 0.125 Ω. The ambient temperature was set to 25 °C.

The power losses of the switches in the first switching state of the six-step switching sequence were assessed first. The characterized power losses could then be applied to the other switching steps. The first switching state involved three inverter switches: S_1_, S_2_, and S_4_. Basically, S_1_ was a PWM power MOSFET in which the switching signal was discontinuously “on” (i.e., PWM “on”) with a duty cycle of 50%, S_2_ was an FWD switch in the commutation step, and S_4_ was a commutation power MOSFET in which the switching signal was continuously “on”. Accordingly, switching loss occurred only in S_1_ (power MOSFET) and S_2_ (diode), whereas conduction loss took place in all these three inverter switches. This switching state comprised two current loops during a single PWM cycle: PWM “on” and PWM “off”. The parasitic inductances involved in the PWM “on” loop were *L*_1_*, L*_8_*, L*_9_*, L*_10_*, L*_11_*, L*_8_*, L*_9_*, L*_5_, and *L*_7_ and those in the PWM “off” loop were *L*_11_*, L*_8_*, L*_9_*, L*_4_*, L*_9_*, L*_8_*,* and *L*_10_. Next, circuit simulations of the power MOSFET inverter during the first switching state at eight different temperatures, i.e., 25, 50, 75, 100, 125, 150, 175, and 200 °C, were performed with the detailed circuit simulation model shown in Figure 2a. The calculated power losses of the inverter switches, S_1_, S_2_, and S_4_, in the first switching state as a function of temperature are displayed in Figure 9a. In the figure, the legend of the light blue solid line with rectangle symbols, i.e., “Diode power loss”, indicates the sum of the switching and conduction losses of the FWD switch. The results demonstrate that the switching and conduction losses of S_1_, the diode power loss of S_2_ (FWD), and the conduction loss of S_4_ in the first switching state were around 10.8, 19.8, 51.6, and 17.7 W at 25 °C and increased or decreased to around 11.2, 25.7, 41.3, and 30.2 W at 200 °C. Specifically, in contrast to the diode power loss of S_2_, the switching and conduction losses of the S_1_ and S_4_ switches tended to increase with increasing temperature. Noticeably, the diode conduction loss (S_2_) showed a relatively strong and negative temperature coefficient, predominantly due to the diode characteristics shown in Figure 3d, where an increased temperature revealed a reduced drain-source voltage under the same drain-source current, thereby leading to a decreased conduction loss. Furthermore, it is interesting to note that temperature had a much smaller impact on the switching loss as compared to the conduction loss, that the switching loss of S_1_ was much smaller than its conduction loss, and that the diode power loss outperformed the PWM (S_1_) and commutation (S_4_) power MOSFET modules. 

The total power loss of the inverter in the first switching state increased from about 99.9 W at 25 °C to about 108.4 W at 200 °C. The insignificant increase in the total power loss was mainly due to the negative temperature coefficient of the diode power loss. The total power loss at 25 °C was used as the initial power level for the development of the Foster thermal network model. Similarly, the power losses of these inverter switches at the other five switching states of the six-step switching sequence could also be derived in the temperature range of 25–200 °C. The calculated power losses during one PWM six-step commutation cycle at 25 and 200 °C are presented in Table 3 and Table 4. It can be clearly seen that each inverter switch conducted for 120 electrical degrees in each periodic cycle for the common 120-degree square-wave commutation. For each inverter switch at each temperature, the power losses that occurred in the six switching states were averaged, and the results at 25 and 200 °C are also listed in the tables; with these, the equivalent electrical resistances (*R*_1_*–R*_6_) can be derived and the results at 25 and 200 °C are also demonstrated in the tables. The average power loss across one PWM six-step commutation cycle was used in the subsequent ETCA analysis. 

Using the proposed ETCA approach, the transient maximum junction temperature profiles of the six inverter switches under natural convection over a time span of one second were calculated and compared with those of the standard ETCA approach. Two examples of the results associated with the inverter switches S_3_ and S_4_ are shown in Figure 9b. The reason for simply conducting the one-second test was that it is very difficult to perform the standard ETCA analysis for a longer period or to solve for the steady-state solution; hence, the more feasible ETCA approach was proposed. Evidently, there was a close agreement between them, suggesting the effectiveness of the proposed analysis approach. The calculated transient maximum junction temperature profiles of these inverter switches using the proposed ETCA are shown in Figure 10a for the time interval [0, 12000 s], and the corresponding temperature distributions in the power MOSFET chips of the inverter at the end of the simulation (t = 12,000 s) are illustrated in Figure 10b. Figure 10a reveals that the maximum junction temperatures of the power MOSFET chips would approach a steady state at around 4000 s. The maximum steady-state junction temperatures of the lower-side switches (namely S_2_, S_4_, and S_6_) would be reached around 160 °C, while those of the upper-side switches (i.e., S_1_, S_3_, and S_5_) would be reached at about 152 °C. These maximum junction temperatures exceed the maximum junction temperature rating of 150 °C and would not be permitted for device reliability and performance concerns. Active convection cooling, such as fans, or passive convection cooling, such as heat sinks and heat pipes, can be effective means to reduce the device junction temperature.

The predicted maximum device junction temperatures of the three-phase inverter during the six-step operation unfavorably exceed the maximum junction temperature rating of 150 °C. The issue can be solved by active cooling with forced air. The CFD analysis of forced convection heat transfer was carried out with two wind speeds, 1.5 and 3.0 (m/s). The direction of the air flow was set to be horizontal, i.e., the x-axis in Figure 6. It can be noted that the Foster thermal network model derived above is no longer be applicable in this ETCA analysis due to its having different transient thermal impedance responses. Thus, a new Foster thermal network model was constructed. The total power loss at 25 °C, i.e., 99.9 W, was also applied as the initial power level to create the Foster thermal network model. The analysis results are displayed in Figure 11. For comparison, the natural convection result (i.e., wind speed = 0 m/s in the figure) is also demonstrated. The device junction temperature under natural convection is around 160 °C, and it is greatly reduced down to about 135 °C under forced convection with an air flow rate of 3 m/s. In addition, the increase in the air flow rate elevates the heat removal performance and thus lowers the device junction temperature.

## 7. Conclusions

This article presented an effective and efficient ETCA approach to characterize the ET coupled behavior of power systems under natural and forced convection during load cycles, which cannot be achieved using the conventional standard ETCA approach. The effect of temperature on the power losses and the influence of parasitics on the switching transients and power losses were all taken into account in the investigation. With this approach, the ET performance of a three-phase power MOSFET inverter for brushless DC motor drive under natural and forced convection during load cycles was explored. Additionally, both detailed and simplified circuit models were introduced, where the former was applied to develop the standard ETCA approach as well as the power loss–temperature relationship, while the latter was used to establish the proposed ETCA approach. Moreover, a Foster thermal network model for the three-phase inverter was created using the thermal impedance curves, which were derived through parametric transient CFD thermal analysis. The validity of the developed Foster thermal network model and the proposed ETCA approach was confirmed through the CFD thermal analysis and a standard ETCA approach.

The detailed circuit simulation demonstrated that the power losses (switching and conduction) of the PWM switches (e.g., S_1_ in the first switching state) and the commutation switches (e.g., S_4_ in the first switching state) had a positive temperature correlation while that of the PWM switches and the FWD switches (e.g., S_2_ in the first switching state) had a negative temperature correlation. Moreover, in comparison with the PWM and commutation switches, the FWD not only had the largest power loss but also a relatively strong and negative temperature coefficient. This explains why the total power loss of the inverter would only slightly increase as temperature increases from 25 °C to 200 °C. It was also found that temperature played a much greater role in the conduction loss than the switching loss, and the switching loss of the PWM switches was considerably lower than its conduction loss. 

The proposed ETCA analysis revealed that the maximum junction temperatures of the inverter switches would approach a steady state at around 4000 s, and the lower-side switches (namely S_2_, S_4_, and S_6_) outperformed the upper-side switches (i.e., S_1_, S_3_, and S_5_) in terms of the maximum steady-state junction temperature. Furthermore, these maximum junction temperatures of the inverter switches under natural convection with the specific load condition all exceeded the maximum junction temperature rating, and forced convection cooling with air was judged to be a very effective means to decrease the maximum junction temperatures.

## Figures and Tables

**Figure 1 materials-14-05427-f001:**
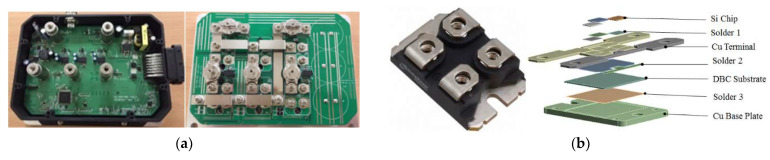
(**a**) Three-phase bridge inverter and (**b**) power MOSFET module and explosive view.

**Figure 2 materials-14-05427-f002:**
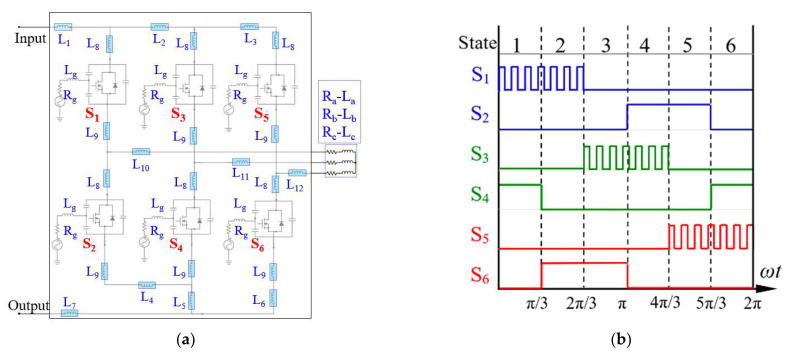
(**a**) Three-phase inverter circuit with parasitic inductances and (**b**) six-step SWPWM signal sequence.

**Figure 3 materials-14-05427-f003:**
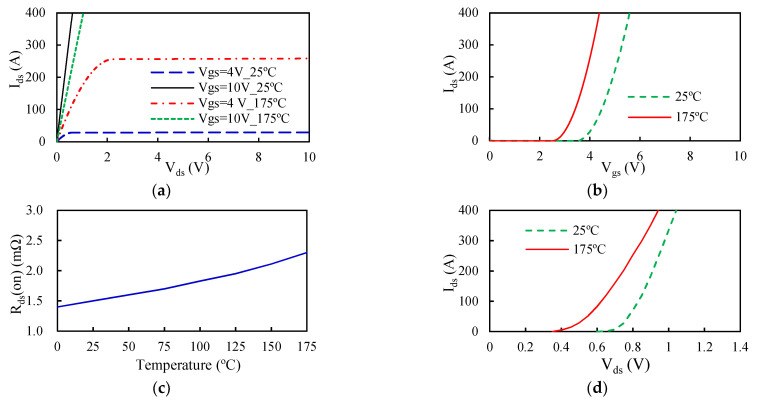
Characteristics of the power MOSFET and body diode presented in the manufacturer’s datasheet and also in [4]: (**a**) power MOSFET output characteristic; (**b**) power MOSFET transfer characteristic; (**c**) power MOSFET on-state resistance; (**d**) diode characteristic.

**Figure 4 materials-14-05427-f004:**
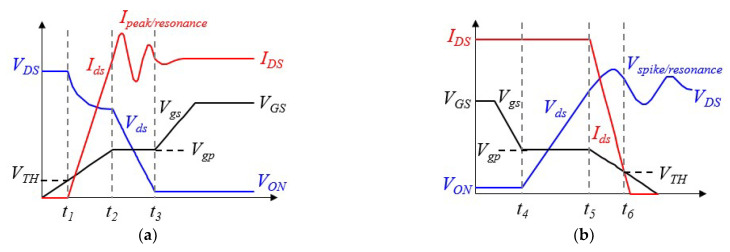
Power MOSFET switching transient: (**a**) turn-on waveform and (**b**) turn-off waveform.

**Figure 5 materials-14-05427-f005:**
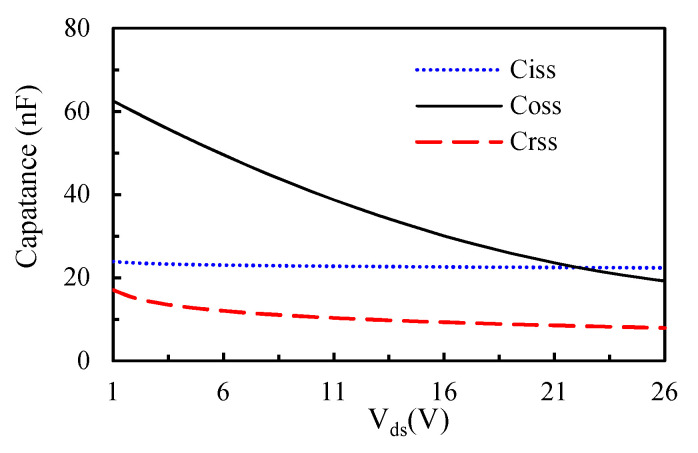
V_ds_-dependent input, output, and reverse transfer capacitances.

**Figure 6 materials-14-05427-f006:**
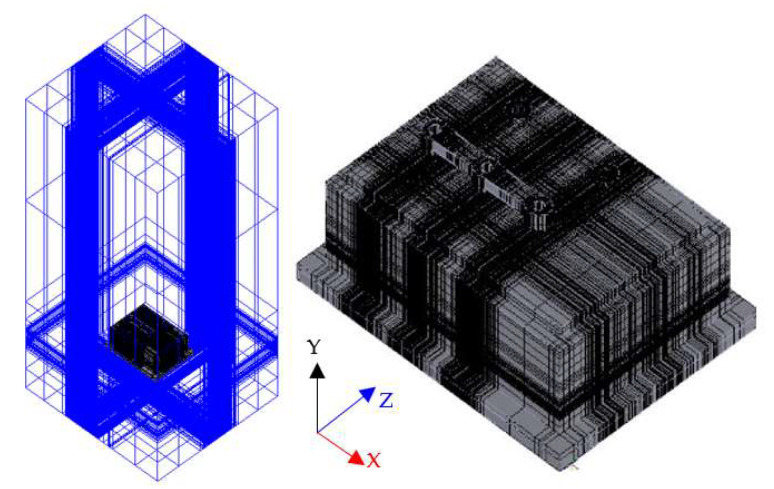
CFD thermal analysis 3D model of the three-phase inverter.

**Figure 7 materials-14-05427-f007:**
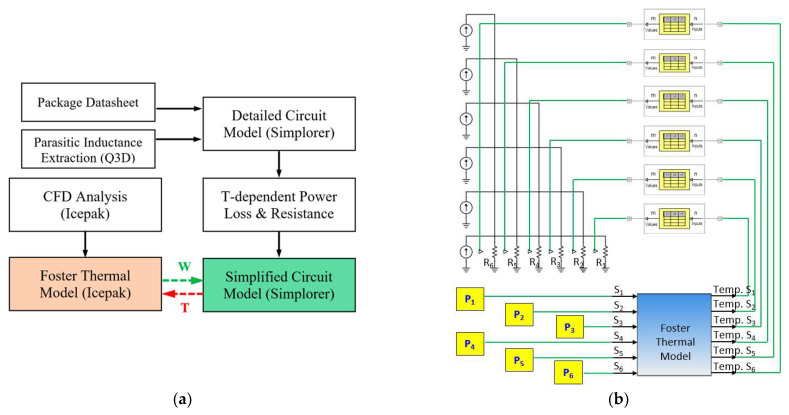
(**a**) Analysis flow of the proposed ETCA model and (**b**) simplified equivalent circuit model.

**Figure 8 materials-14-05427-f008:**
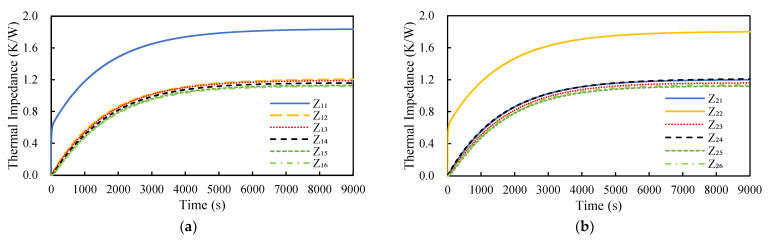
Two examples of the transient thermal impedance curves: (**a**) Z_1*i*_(t), *i* = 1,…6 and (**b**) Z_2*i*_(t), *i* = 1,…6.

**Figure 9 materials-14-05427-f009:**
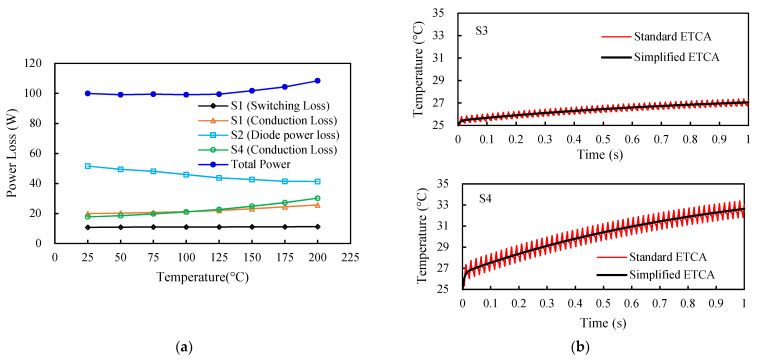
(**a**)Temperature-power loss dependence in the first switching state and (**b**) a comparison of the transient maximum junction temperatures of the switches S_3_ and S_4_ for the standard and proposed ETCAs during a one-second operation.

**Figure 10 materials-14-05427-f010:**
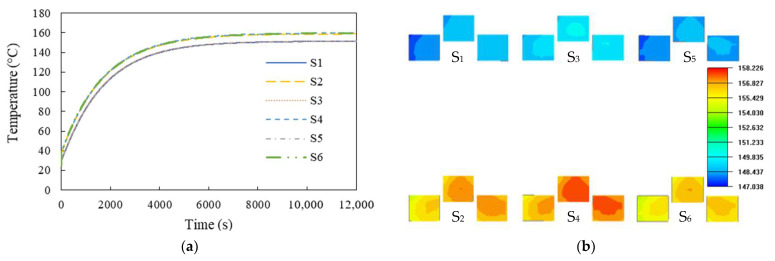
Thermal behavior of the six inverter switches: (**a**) transient maximum junction temperature profiles and (**b**) temperature distribution in the MOSFET chips.

**Figure 11 materials-14-05427-f011:**
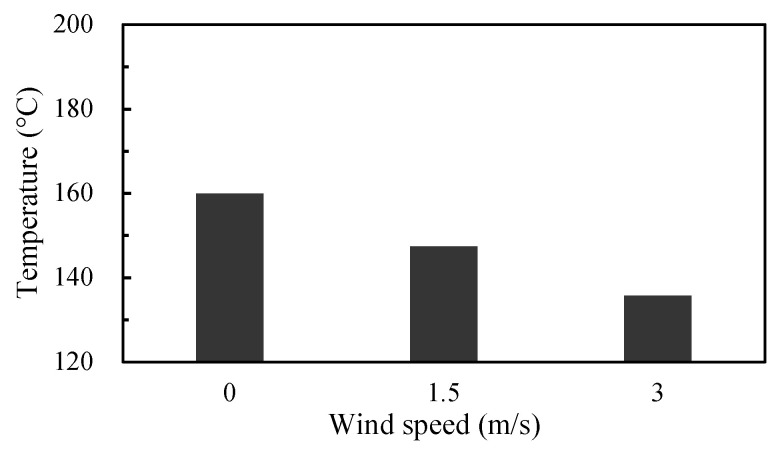
Maximum junction temperature of the inverter switches under forced convection with two different wind speeds.

**Table 1 materials-14-05427-t001:** Curve-fitted resistances and time constants associated with thermal impedances Z_1*i*_ and Z_2*i*_ (*i =* 1,…6).

	Z_11_	Z_12_	Z_13_	Z_14_	Z_15_	Z_16_
R_i_	1.82	1.211	1.199	1.173	1.147	1.137
τ _i_	1151	1605	1629	1678	1730	1749
	Z_21_	Z_22_	Z_23_	Z_24_	Z_25_	Z_26_
R_i_	1.207	1.791	1.175	1.213	1.139	1.153
τ _i_	1600	1253	1677	1604	1745	1707

**Table 2 materials-14-05427-t002:** Comparison of steady-state junction temperatures in the CFD analysis and Foster network (unit: °C).

P^T^(W)	Method	S_1_	S_2_	S_3_	S_4_	S_5_	S_6_
99.9	Foster	149.8	154.7	150.1	155.3	149.1	154.1
	CFD	149.8	154.2	150.0	154.9	149.6	155.0
103.8	Foster	148.6	156.1	150.6	158.2	151.2	153.5
	CFD	153.2	159.6	154.6	161.7	155.9	158.4

**Table 3 materials-14-05427-t003:** Power losses and equivalent resistances of these six inverter switches during one PWM six-step commutation cycle at 25 °C.

—	S_1_	S_3_	S_5_	S_2_	S_4_	S_6_	Total
Step 1	30.62	0.00	0.00	51.56	17.73	0.00	99.92
Step 2	30.62	0.00	0.00	51.56	0.00	17.73	99.92
Step 3	0.00	30.62	0.00	0.00	51.56	17.73	99.92
Step 4	0.00	30.62	0.00	17.73	51.56	0.00	99.92
Step 5	0.00	0.00	30.62	17.73	0.00	51.56	99.92
Step 6	0.00	0.00	30.62	0.00	17.73	51.56	99.92
Average (W)	10.21	10.21	10.21	23.10	23.10	23.10	—
*R_i_* (Ω)	0.00102	0.00102	0.00102	0.00231	0.00231	0.00231	—

**Table 4 materials-14-05427-t004:** Power losses and equivalent resistances of these six inverter switches during one PWM six-step commutation cycle at 200 °C.

—	S_1_	S_3_	S_5_	S_2_	S_4_	S_6_	Total
Step 1	36.90	0.00	0.00	41.31	30.17	0.00	108.38
Step 2	36.90	0.00	0.00	41.31	0.00	30.17	108.38
Step 3	0.00	36.90	0.00	0.00	41.31	30.17	108.38
Step 4	0.00	36.90	0.00	30.17	41.31	0.00	108.38
Step 5	0.00	0.00	36.90	30.17	0.00	41.31	108.38
Step 6	0.00	0.00	36.90	0.00	30.17	41.31	108.38
Average (W)	12.3	12.3	12.3	23.8	23.8	23.8	—
*R_i_* (Ω)	0.00123	0.00123	0.00123	0.00238	0.00238	0.00238	—

## Data Availability

Data sharing not applicable.

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
