# Peer review of "Transient Electro-Thermal Coupled Modeling of Three-Phase Power MOSFET Inverter during Load Cycles"

_materials, 2021, doi:10.3390/ma14185427_

Round 1

Reviewer 1 Report

This manuscript has explored the electrical-thermal behaviors of a three-phase power MOSFET inverter utilizing Foster thermal network model. The electro-thermal coupling analysis (ETCA) approach given by the authors are quite interesting and realistic. This manuscript is recommended for publication after some minor modifications mentioned below.

  1. Figure 2 and 3 looks quite identical to the figures already published in the articles (Journal of Mechanics, 2020, 37, 134–148, 10.1093/jom/ufaa022) presented by the same author. Please clarify the issue.
  2. Please recheck the figure 8 for transient thermal impedance curves.

What is the main question addressed by the research?

The challenge of straight application of Foster resistance-capacitance (RC) thermal networks can be an effective and favored means for junction temperature estimation due to their unparalleled computational efficiency. However,  according to the author the standard ETCA approach mentioned before cannot address difficulties of electrical circuit simulation. Therefore, author has establish a more effective transient ETCA approach that can relieve the challenge. An effective and efficient transient ETCA approach is proposed in their work, which integrates a 3D EM model, a simplified equivalent circuit model, and a Foster thermal network model. According to them this circuit model and thermal model are fully coupled to account for the effect of the temperature on the power losses.The 3D EM model in this manuscript is applied to consider the impact of parasitics on the switching transients and loss. The proposed ETCA approach is demonstrated on the estimation of the electrical-thermal behavior of a voltage source three-phase bridge MOSFET inverter for brushless DC (BLDC) motor drive in natural and forced convection during the six-step operation.

Is it relevant and interesting? How original is the topic?

----- The topic of the manuscript is relevant. Although some of the figures presented here were already published in an earlier publication by the same author. Please see Journal of Mechanics, 2020, 37, 134–148, DOI: 10.1093/jom/ufaa022.

What does it add to the subject area compared with other published material?

----- Previous several works were on a component-level exploration of the switching regarding power loss and thermal behaviors in power modules but in this work on the system level as in three-phase bridge inverters.

Is the paper well written?

----- The article is well written.

Is the text clear and easy to read?

----- The text is clear and easy to understand.

Are the conclusions consistent with the evidence and arguments presented?

----- The conclusion is ok.

Do they address the main question posed?

----- Yes, they have addressed. 

Author Response

This manuscript has explored the electrical-thermal behaviors of a three-phase power MOSFET inverter utilizing Foster thermal network model. The electro-thermal coupling analysis (ETCA) approach given by the authors are quite interesting and realistic. This manuscript is recommended for publication after some minor modifications mentioned below.

Response: The authors would like to express our sincere thanks to the reviewer for the support of this work and also the provision of valuable comments and suggestions for this work. The reply to the questions or comments of the reviewer is given below.

  1. Figure 2 and 3 looks quite identical to the figures already published in the articles (Journal of Mechanics, 2020, 37, 134–148, 10.1093/jom/ufaa022) presented by the same author. Please clarify the issue.

Response: The authors truly appreciate the question. Indeed, Figure 2 and 3 are similar to those shown in the authors’ previously published article (Journal of Mechanics, 2020, 37, 134–148, 10.1093/jom/ufaa022). It is, however, noted that Figure 2 simply demonstrates the equivalent electrical circuit of the three-phase inverter operated in six-step 120o square wave PWM (SWPWM) commutation, and Figure 3 reveals the transport characteristics of the SOT-227 packaged power MOSFET, such as transfer characteristics and output characteristics. They are nothing but the circuit model of the test vehicle and the corresponding input electrical properties for the circuit simulation. That paper aimed to conduct a numerical and experimental characterization of the switching transients and power losses of Si power MOSFET module and three-phase MOSFET inverter during load cycles. Though the research vehicle of these two studies is nearly the same, their aims are totally distinct. The present study can be considered as an extension of the previous study.

  1. Please recheck the figure 8 for transient thermal impedance curves.

Response: Thank you very much for the question. Figures 8(a) and 8(b) demonstrate the thermal impedance curves of Z1i(t) (i=1,….6) and Z2i(t) (i=1,….6), respectively. Indeed, it is a bit confusing in the “key to the symbols”, namely S1, S2, ….and S6. They are changed to Z11, Z12, Z13, Z14, Z15, and Z16 in Fig. 8(a) and Z21, Z22, Z23, Z24, Z25, and Z26 in Fig. 8(b) in the revised manuscript.

Reviewer 2 Report

This paper focus on a transient electro-thermal coupling analysis to analyze the electrical and thermal behaviors of a three-phase power MOSFET inverter for a BLDC motor drive in natural and forced convection, assuming six-step operation. The coupled analysis a  3D electromagnetic simulation for parasitic parameters extraction to determine power losses, and a thermal network model to predict the temperature of the junction.

The paper deals with an interesting topic, although some minor and major observations must be addressed before a final decision can be taken.

1.- English grammar and style require a moderate revision.

2.- Section 1 requires to highlight in deep the main contributions and novelties of the paper, if possible in list form, comparing with the state of the art research. As redacted, it seems that there is a lack of significant contributions, so that the authors must prove that the technical contributions are not incremental, and the proposed methodology is novel, and the overall impact is significant enough.

3.- Section 4.3. It is not clear how the authors link the C and R elements of the thermal network with (21).

The Reviewer suggests revising the work based on the suggestions above in order to improve its readability, scientific interest and quality.

Author Response

Dear Reviewer: Please see the attachment. Thanks.

Round 2

Reviewer 2 Report

The authors have replied my questions

Author Response

Comments and Suggestions for Authors :  The authors have replied my questions

Response: 

The authors would like to express our sincere thanks to the reviewer for the support of this work.